

# Seed priming with essential oils for sustainable wheat agriculture in semi-arid region

Muhammet Çağrı Oğuz, Ezgi Oğuz and Mustafa Güler

Department of Field Crops, Ankara University, Ankara, Turkey

## ABSTRACT

Drought is one of the major constraints to global crop production. A number of sustainable systems have focused on the development of environmentally friendly innovative biotechnological interventions to prevent yield losses. The use of essential oils as a seed priming agent can make an important contribution as a natural stimulant in increasing drought stress tolerance. This study focuses on the effects of seeds coated with different doses ($D_0$ (0%), $D_1$ (0.01%), $D_2$ (0.05%), $D_3$ (0.10%) and $D_4$ (0.25%)) of sage, rosemary and lavender essential oils on wheat germination, seedling establishment and yield parameters. Turkey's local wheat genotype Köse was used as plant material. The impact of the seed priming on germination rate, coleoptile length, shoot length, root length, shoot fresh and dry weight, root fresh and dry weight, relative water content (RWC), proline, and chlorophyll contents was assessed in laboratory experiments. In addition, the effect of essential oil types on yield parameters and agronomic components (plant height, spike height, number of grains per spike, grain yield per spike, grain yield per unit area, thousand-grain weight) was evaluated in a field experiment during the 2019–2020 crop seasons in a semi-arid climate. According to laboratory results, the highest germination rate among all treatment doses was determined in the $D_2$ treatment (rosemary 93.30%, sage 94.00% and lavender 92.50%), while the lowest germination rates for all essential oil types were determined in the $D_4$ treatment (rosemary 41.70%, sage 40.90% and lavender 40.90%). Increasing treatment doses showed a similar suppressive effect on the other parameters. In the field experiment, the highest grain yield (256.52 kg/da) and thousand-grain weight (43.30 g) were determined in the rosemary treatment. However, the priming treatment has an insignificant on the number of grains per spike and the spike length. The light of these results, the effects of essential oil types and doses on yield parameters were discussed. The findings highlight the importance of using essential oils in seed priming methods for sustainable agricultural practices.

## INTRODUCTION

The importance of wheat production is increasing day-by-day to meet the nutritional needs of the growing population around the world. According to the wheat production

Corresponding author
Muhammet Çağrı Oğuz,
m.cagrioguz@gmail.com

data for 2020, 765 million tons of wheat were produced in 223.9 million ha of agricultural land and 3.4 tons of yield per ha was obtained (*USDA, 2021*). According to the report of the International Grains Council (IGC), wheat production is expected to be close to 787 million tons in the 2021/2022 production period. It is stated that in addition to this expected increase in production, the estimated amount of consumption will increase similarly (*IGC, 2022*).

Intensive cultural and chemical processes to get more yields from agricultural production injure the ecological cycle and nature (*Souri, 2016*; *Kisiriko et al., 2021*). Global climate change, which is a result of the deteriorating balance of nature, causes problems such as irregularity in temperatures, anomalies in seasonal processes and drought (*Malhi, Kaur & Kaushik, 2021*). Drought is one of the main factors of production restrictions that affect plant growth, development and yield. The increasing drought effect due to climate change in many arid and semi-arid regions has limited agricultural activities (*Trenberth, Dai & van der Schrier, 2014*; *Golla, 2021*). Approximately 45% of wheat production globally is affected by drought (*Afzal et al., 2015*). A number of sustainable strategies have focused on the development of agro-ecosystems to prevent yield losses and increase drought tolerance, the efficient use of water resources, and environmentally friendly innovative biotechnological interventions to end the use of more chemical fertilizers and pesticides. In this context, the quest for environmentally friendly, sustainable methods and approaches has continued to minimize the increasing damage of climate change (*Yakhin et al., 2017*; *Joshi et al., 2017*; *Wani et al., 2018*; *Del Buono, 2020*). For this purpose, biostimulants and bio-protectants have been proposed as potential agents to promote plant growth and increase yield (*Bulgari, Franzoni & Ferrante, 2019*; *Singh et al., 2020*).

Today, the usage areas of plants have expanded with increasing biotechnological methods. Antibacterial, antifungal and antioxidant properties of essential oils (EO) obtained from medicinal and aromatic plants are promising approaches for extending product shelf life, biological control against diseases and pests, and increasing plant stress tolerance to abiotic stress factors (*Cuadrado et al., 2019*; *Kahramanoglu & Usanmaz, 2021*; *Kesraoui et al., 2022*). Especially their easy metabolism has paved the way for their use instead of synthetic chemicals in agricultural areas (*Belasli et al., 2020*). Bio-based compounds such as phenols, flavonoids, quinones, tannins, alkaloids, saponins and sterols found in plant extracts and essential oils are an alternative to chemical drugs used to control fungal and bacterial pathogens (*El-Awady, 2019*). The usage of essential oils with seed coating and priming technologies could make an important contribution as a natural stimulant in increasing biotic and abiotic stress tolerance (*Tavares et al., 2013*; *Lutts et al., 2016*; *Ben-Jabeur et al., 2019*; *Mrid et al., 2021*).

Scientific studies mostly focused on the antifungal, antibacterial and insecticidal effects of plant essential oils (*Mancini & Romanazzi, 2014*; *Belasli et al., 2020*; *Moumni et al., 2021*; *Kesraoui et al., 2022*). Besides, some studies have determined the effects of essential oils on germination and seedling formation in abiotic stress tolerance (*Yavaş, 2010*; *Atak, Mavi & Uremis, 2016*; *Bingöl & Battal, 2017*; *Souri & Bakhtiarizade, 2019*; *Binbir et al., 2019*; *Ben-Jabeur et al., 2019*; *Săndulescu, Manole & Stavrescu-Bedıvan, 2020*). However, these studies are generally limited to laboratory experiments. There is still limited

information about the effects of seed priming with essential oils on yield and physiological parameters in field trials. Therefore, testing seed priming methods in field trials will significantly contribute to scientific studies and farmer needs (*El Boukhari et al., 2020*).

The prerequisite for well plant growth in drought conditions is germination. Early germination and strong winter penetration of the plant significantly increase its yield performance. Pretreatment of seeds with essential oil or extracts obtained from plants shows a biostimulant effect to increase tolerance to drought stress (*Farooq et al., 2018*; *Ben-Jabeur et al., 2022*). Seed preparation techniques have a significant effect on the metabolic, biochemical and enzymatic activities of the seed (*Nile et al., 2022*). In this way, seed priming allows early germination and strong seedling formation in arid conditions (*Raj & Raj, 2019*; *Zulfiqar, 2021*). Such treatments are seen among the promising approaches for abiotic and biotic stress tolerance in arid and semi-arid regions (*Sharma et al., 2014*; *Ben-Jabeur et al., 2019*).

This study investigated the usability of essential oils of rosemary (*Rosmarinus officinalis* L.), sage (*Salvia officinalis* L.) and lavender (*Lavandula x intermedia* L.) as seed priming material in wheat. We hypothesized that essential oil treatment on seed has a positive effect on germination and seedling in wheat. In this context, we focus to determine the effect of essential oil types and doses. For this purpose, laboratory experiments were carried out in a growth chamber with constant temperature, regular light and constant humidity for 2 weeks. In the second hypothesis created according to laboratory experiment results, we accepted that seed priming with essential oil could reduce the effect of possible drought stress in the semi-arid region and have a positive effect on yield in sustainable wheat agriculture. The field trial allowed the seed priming studies with essential oils not to be limited to laboratory trials. To the best of our knowledge, this is one of the rare and advanced studies in which essential oil seed priming was investigated under laboratory and field conditions for sustainable wheat agriculture.

# MATERIALS AND METHODS

## Plant materials

The Köse wheat used in the experiments were obtained from the Ankara Yenimahalle Campus Seed Bank of the Ministry of Agriculture and Forestry. Köse wheat (*Triticum aestivum* L. ssp. Vulgare Vill. v. delfii Körn) is the local wheat variety in Turkey. Köse wheat is tolerant to cold and drought stress (*Kan et al., 2015*). Rosemary (*Rosmarinus officinalis* L.), sage (*Salvia officinalis* L.) and lavender (*Lavandula x intermedia* L.) plants were obtained from the field of Ankara University Field Crops Department.

## Extraction method of essential oils

The leaves of rosemary and sage plants and the flowers of the lavender plants were collected and dried at room temperature and in the dark. Dried plant parts were cut into 1 cm small pieces. Essential oil extraction of dry plant parts was obtained by steam distillation method. According to the *European Pharmacopoeia Method (1996)*, freshly chopped plant material (250 g) was inserted into the extraction vessel of the Clevenger apparatus with 2,000 ml of distilled water and extracted for 3 h. The stock essential oils

**Table 1 The components of essential oils of *Rosmarinus officinalis* L., *Salvia officinalis* L. and *Lavandula* x *intermedia* L.**

| Salvia officinalis L. | | Rosmarinus officinalis L. | | Lavandula x intermedia L | |
|---|---|---|---|---|---|
| Components | (%) | Components | (%) | Components | (%) |
| α-pinene | 3.56 | α-pinene | 10.11 | α-pinene | 0.13 |
| Camphene | 3.84 | Camphene | 6.71 | Camphene | 0.29 |
| β-pinene | 1.44 | β-pinene | 2.12 | Myrcene | 0.69 |
| Myrcene | 3.75 | Myrcene | 0.98 | p-cymene | 0.10 |
| γ-terpinene | 1.83 | α-terpinene | 1.13 | Limonene | 1.75 |
| p-cymene | 1.29 | Limonene | 2.88 | Eucalyptol | 5.49 |
| 1.8-cineole | 16.67 | 1.8-cineole | 35.8 | γ-pinene | 1.25 |
| β-ocimine | 3.58 | γ-terpinene | 0.78 | δ-3-carene | 0.77 |
| Terpinolene | 1.47 | p-cymene | 3.23 | Linalool | 28.64 |
| α-thujone | 18.23 | Terpinolene | 0.51 | Octenol | 0.25 |
| β-thjone | 6.76 | Camphor | 16.41 | Camphor | 6.97 |
| Camphor | 5.71 | Linalool | 1.09 | Isoborneol | 4.75 |
| Borneol | 3.69 | Bornyl acetate | 0.92 | α-terpineol | 3.55 |
| Bornyl acetate | 1.20 | Caryophyllene | 1.57 | Nerol | 0.84 |
| Trans-caryophyllene | 9.19 | Terpinenol | 1.43 | Linalyl acetate | 24.73 |
| α-humulene | 9.41 | Isoborneol | 1.01 | Lavandulyl acetate | 3.67 |
| Other | 8.38 | α-terpineol | 7.10 | Geranyl acetate | 0.94 |
| | | Borneol | 6.22 | Cis-geraniol | 1.79 |
| | | | | Thujopsene | 2.00 |
| | | | | β-caryophyllene | 0.61 |
| | | | | Other | 10.79 |

obtained as a result of distillation were stored in amber-colored tubes at 4 °C in the refrigerator.

## GC-MS analysis of essential oils

Analysis of essential oils in the experiments was performed using gas chromatography (GC) coupled to a mass selective detector equipped with an HP-5MS (cross-linked 5%) (*Dadalioğlu & Evrendilek, 2004*). Essential oil samples were passed through the capillary column (50 m × 0.32 mm × 1.2 μm). Helium gas was used as carrier gas. The flow rate is set at 10 psi. The column temperature was 60 °C initially and reached 220 °C with a 2 °C increase per minute. A total of 20 min at 220 °C has been fixed for a period of time. The components of essential oils of rosemary, sage and lavender are given in Table 1.

## Preparation of essential oil solutions

The stock essential oils were dissolved by adding a few drops of Tween 20®. The essential oils used in the experiments were diluted with sterile distilled water from the dissolved oil solution and adjusted to four different doses (0.01%, 0.05%, 0.10% and 0.25%). The control treatment was prepared by adding an equal amount of tween 20 to the distilled water.

Essential oils used in the experiment consists of five different treatment dose as $D_0$ (control 0%), $D_1$ (0.01%), $D_2$ (0.05%), $D_3$ (0.10%), and $D_4$ (0.25%). The image of the essential oil stock and the essential oil solutions obtained by dilution is given in the Fig. S2. The prepared solutions were stored at 4 °C dark conditions until used.

## Seed sterilization and priming treatments in the laboratory experiment

Surface sterilization of wheat seed was done with NaClO (sodium hypochlorite) before essential oil priming. The sterilization method was modified from *Hayta et al. (2021)*. The wheat seeds were washed in 10% NaClO for 10 min with a magnetic stirrer, and then rinsed in sterile distilled water three times for 5 min. Sterilized seeds were primed with diluted essential oils for 12 h in closed petri dishes and in dark conditions. Primed seeds removed from the solutions after 12 h were sown on sterile germination papers in 10 cm glass petri dishes. Afterwards, petri dishes were covered with stretch film.

Laboratory experiments were carried out in five replicates representing doses of essential oil species and ten seeds in each replicate according to the random plot design. Laboratory experiments were maintained under controlled conditions in constant temperature (25 ± 2 °C), regular light (16 h light and 8 h dark) and constant humidity (50%) growth chambers. Germination and seedling development were achieved in the growth chambers for 2 weeks. At the end of 2 weeks, physiological and biochemical measurements were performed on ten plants randomly selected from each repetition of essential oil types and doses.

## Physiological measurements in laboratory experiment

Germination rate, coleoptile length, shoot length, root length, shoot fresh and dry weight, root fresh and dry weight were measured on the 14th day of priming treatment with essential oils (*Zencirci et al., 2019*). Relative water content (RWC) was determined by drying the leaf sample at 105 °C for 30 min and then at 70 °C until a constant weight was reached. After cooling to room temperature, the samples were weighed and the leaf water content was calculated according to the specified formula RWC (%) = $[(FW − DW)/FW] \times 100$ (*Deef, 2007*) (FW, fresh weight; DW, dry weight).

## Proline analysis

The proline content of leaf tissues was determined spectrophotometrically according to the method determined by *Bates, Waldren & Teare (1973)*. A total of 5 g of leaf tissue was treated with 3% sulfosalicylic acid. A total of 2 ml of the obtained plant extract was taken and 2 ml of acid ninhidrin and 2 ml of glacial acetic acid were added to it, and it was kept in a 100 °C water bath for 1 h. Then, 4 ml of toluene was added to each of the samples in the tubes, which were kept in ice for 5 min, and measurements were made at 520 nm in the spectrophotometer.

## Chlorophyll contents analysis

The total chlorophyll contents of seedlings were determined by the *Wellburn (1994)* method. Accordingly, 50 mg of green leaf tissue was placed in 3 ml of methanol and kept at 23 °C for 2 h in the dark. Optical density (OD) was determined by measuring 1.5 ml of

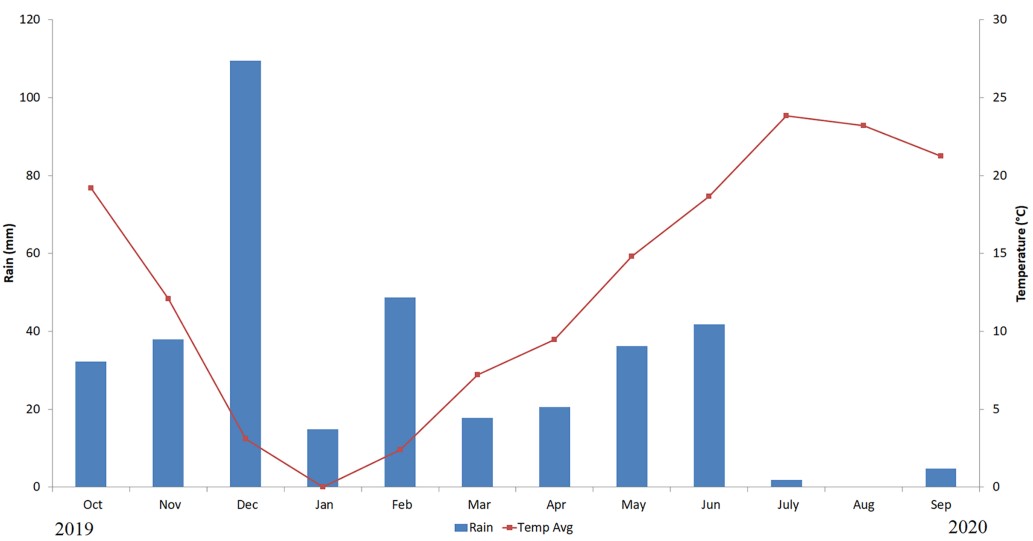

**Figure 1 Temperature and precipitation data for the field experiment 2019–2020.**

liquid from chlorophyll dissolved in methanol using a spectrophotometer at 650 and 665 nm. With the obtained OD values, the amounts of total chlorophyll content were determined according to the formulas below.

Total chlorophyll content $(\mu g/g) = [25.8 \times A650 - 4 \times A665] \times 3/0.05$

## Field trial and the experimental design

In laboratory experiments, the effect of essential oil types and doses and their effects on physiological (germination rate, coleoptile length, shoot length, root length, shoot fresh and dry weight, root fresh and dry weight, RWC) and biochemical (proline and chlorophyll content) measurements were determined through a two-factor analysis of variance. Field trials were carried out with seeds primed with the best essential oil dose determined according to the laboratory results. In the field trial, the effect of seed priming on plant number (PN), plant length (PH), spike length (SL), number of grains per spike (SGN), grain yield per spike (SGY), grain yield per unit area (GY) and thousand-grain weight (TGW) were determined by one-factor analysis of variance. All field trials were carried out in the field of Ankara University Field Crops Department. The research site is located at an altitude of 870 m above sea level, between 39° 57′ north latitude and 32° 51′ east longitude. The experiment was carried out during the 2019–2020 production seasons. Precipitation and temperature data for 2019–2020 are given in Fig. 1.

A proper sowing bed has been prepared before sowing. The field soil was ploughed twice with a plow at a depth of 20–25 cm before sowing. Field trials were established according to the randomized block design. For each essential oil type, the trials were carried out in three replicates. Each plot consisted of 10 rows of 2 m in length, with a row spacing of 0.25 m. Sowing was carried out in the first week of October 2019.

The experiment was carried out under precipitation-based production conditions. Wheat seeds have been sterilized before priming. According to the laboratory experiment results, the essential oil was applied at the rate of $D_2$ (0.05%), the most suitable average value determined in all parameters. Sterilized seeds were treated with $D_2$ essential oils of rosemary, sage and lavender for 12 h at room temperature. Seeds were prepared as 500 seeds per $m^2$ for each trial plot. A total of 10 ml of $D_2$ rosemary, sage and lavender essential oil solutions were added to the prepared seeds, and they were placed in plastic bags and sealed. At the end of 12 h, the seeds were planted in the plots prepared in the field. Any organic or synthetic fertilizers and pesticides were not applied to the plants before and after planting. Weeds in the plot areas have been cleaned with hand tools. Precipitation-based production was carried out without irrigation, including during the planting period.

The parameters related to the yield examined in the field trials were carried out on 10 randomly selected plants. The investigated characteristics are the number of plants per square meter, plant height (cm), spike height (cm), number of grains per spike, grain yield per spike (g), grain yield per unit area (kg/da), thousand-grain weight (g).

## Statistical analysis

The data obtained for all the results that were measured and observed in the experiment were subjected to analysis of variance (ANOVA) in the IBM SPSS Statistics 22.0 program. The laboratory experiment was two factorial Split Plot Design. The types of essential oils were the first factor, assigned to the main plots, essential oil doses were the second factor allocated to the subplots (split plots). The field experiment was one factorial completely randomized design, where a comparison was noted among different types of essential oils. The differences between the means in laboratory and field trials were determined using the Duncan test at 0.01 level (*Snedecor & Cochran, 1967*).

# RESULTS

## Physiological results in the laboratory experiment

Wheat germination rates in $D_0$ (control), $D_1$, $D_2$, $D_3$ and $D_4$ treatments prepared with essential oils of rosemary, sage and lavender are indicated in Fig. 2A. The highest germination rate among all treatment doses was determined in the $D_2$ treatment (rosemary 93.30%, sage 94.00% and lavender 92.40%). There was no statistically significant difference in germination between $D_0$ and $D_1$ EO treatments (Fig. 2A). On the other hand, germination rates decreased in $D_3$ and $D_4$ EO treatments. The lowest germination rates for all three essential oil types were determined at the $D_4$ treatment (rosemary 41.70%, sage 40.90% and lavender 40.90%) (Fig. 2A). The most effective essential oil type on germination rate is sage, followed by rosemary and lavender (Fig. 3A).

The effect of rosemary essential oil treatment on coleoptile length is shown in Fig. 4A, sage in Fig. 4B and lavender Fig. 4C. The highest coleoptile length was determined at the $D_1$ and $D_2$ treatment doses (mean 6.40 and 6.36 cm, respectively) (Fig. 2B). In the experiments, $D_1$ and $D_2$ treatments showed a statistically significant increase in coleoptile length compared to the $D_0$ treatment, while $D_3$ and $D_4$ treatments showed a suppressive

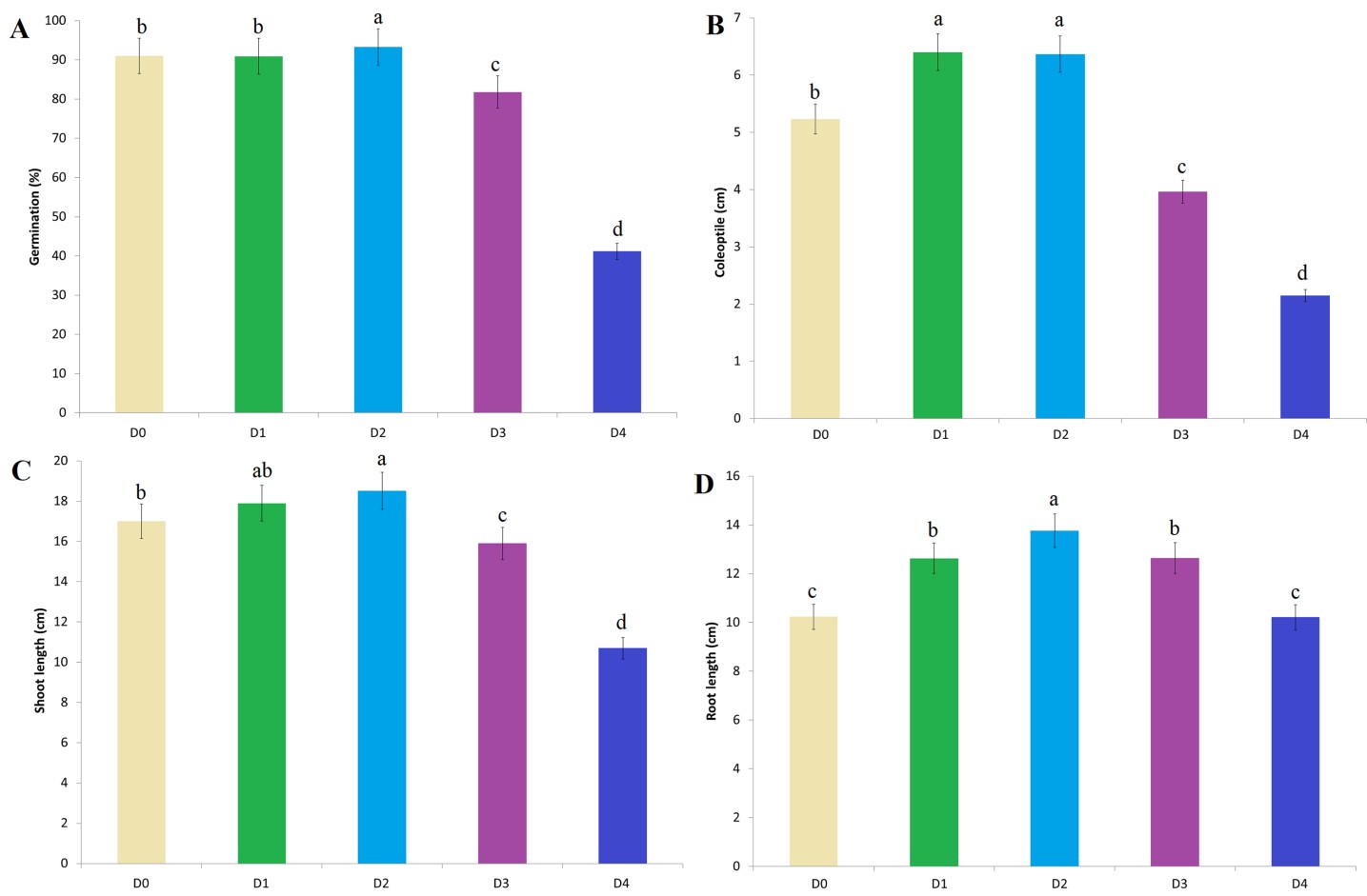

**Figure 2 The effect of essential oil treatment doses on physiological parameters.** (A) Germination rate. (B) Coleoptile length. (C) Shoot length. (D) Root length. Treatment doses represent D0: control, D1: 0.01%, D2: 0.05%, D3: 0.10%, D4: 0.25%. The difference between the means shown with different letters is significant according to the Duncan test at the 0.01 level.

effect on coleoptile length (Fig. 2B) (Figs. 4A–4C). The most effective essential oil on coleoptile length is sage (Fig. 3B). There is no significant difference between the effects of rosemary and lavender essential oil on coleoptile length (Fig. 3B).

A significant increase in plant shoot length was observed with essential oil priming compared to the control (Fig. 2C). The best shoot length was measured in the $D_2$ treatment (mean 18.51 cm), although the difference with the $D_1$ (mean 17.90 cm) was insignificant. The lowest shoot length compared to the $D_0$ control was determined in the $D_4$ treatment (rosemary, 11.00 cm, sage 9.40 cm, lavender 11.70 cm) (Fig. 2C). Compared to sage and lavender, the most effective essential oil on shoot length is rosemary (Fig. 3C).

The best root length was measured in the $D_2$ treatment compared to the $D_0$ control treatment. The difference between the $D_1$ and $D_3$ treatments was found to be insignificant (Fig. 2D). Similarly, the difference between root length in $D_0$ and $D_4$ treatments are statistically insignificant. The root length difference between rosemary and sage was statistically insignificant (Fig. 3D). Both essential oils showed a significant positive effect on root length compared to lavender.

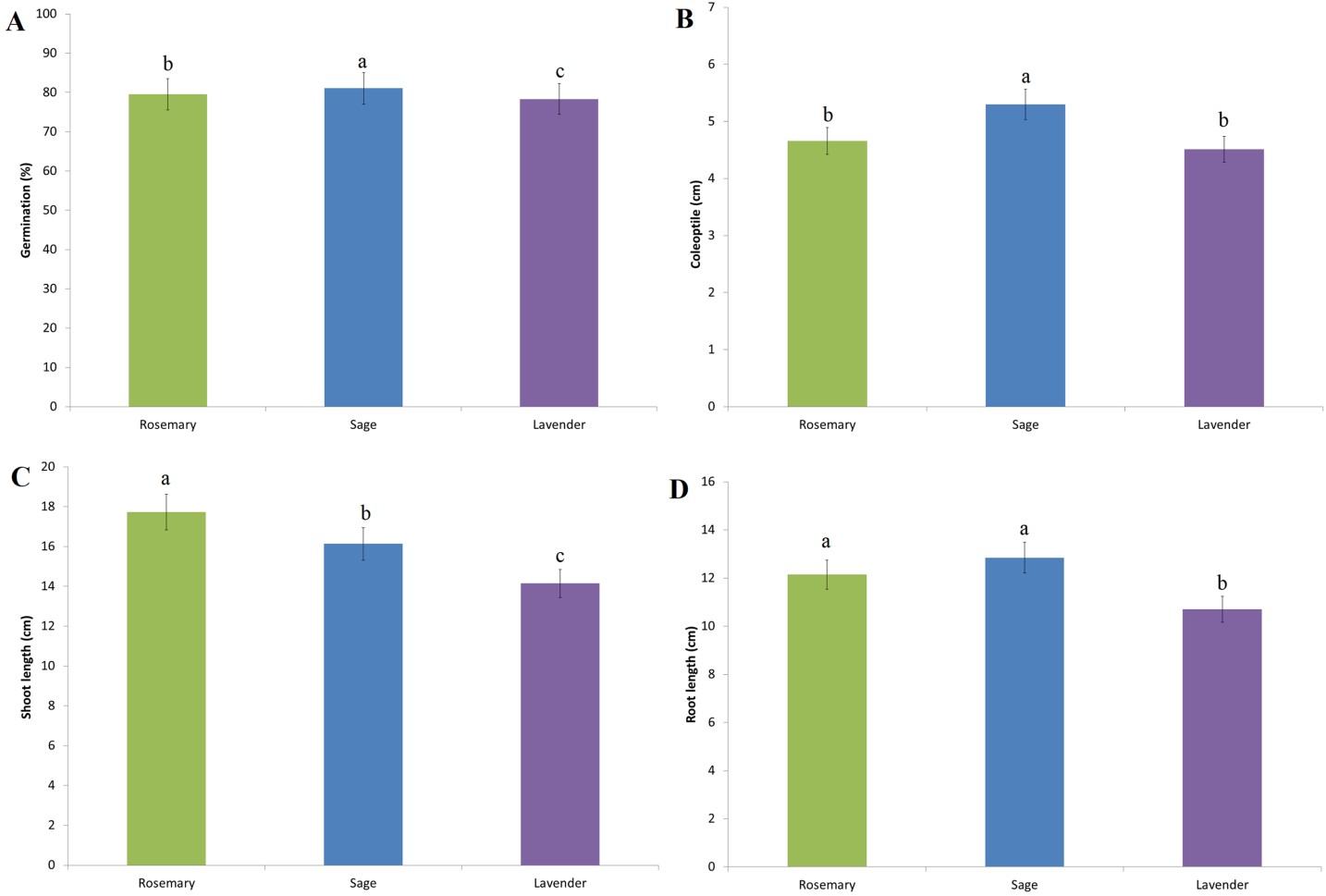

**Figure 3 The effect of essential oil types on physiological parameters.** (A) Germination rate. (B) Coleoptile length. (C) Shoot length. (D) Root length. Treatment doses represent D0: control, D1: 0.01%, D2: 0.05%, D3: 0.10%, D4: 0.25%. The difference between the means shown with different letters is significant according to the Duncan test at the 0.01 level.

## Biochemical analysis results in the laboratory experiment

The effects of EO seed treatments on the proline content of the plant were determined during the seedling period (Fig. 5A). Compared with the $D_0$ treatment, the $D_1$ treatment caused a reduction in proline accumulation. A similar reduction was found at other treatment doses. The most significant reduction in proline accumulation was determined at the $D_2$ treatment dose (Fig. 5A). The type of essential oil that caused the most significant reduction in proline accumulation was sage compared to rosemary and lavender (Fig. 6A).

RWC contents were highest in $D_2$ (72.90%) compared to the $D_0$ treatment (69.88%). The RWC value of 70.46% determined at the $D_1$ treatment dose was significant compared to the $D_0$ control. The RWC values of 69.45% and 67.62% determined in the $D_3$ and $D_4$ treatment doses are the doses in which the lowest values were determined compared to the control $D_0$ treatment (Fig. 5B). The essential oil type that has the most important effect on RWC values is rosemary. The difference between lavender and sage essential oil is also important (Fig. 6B).

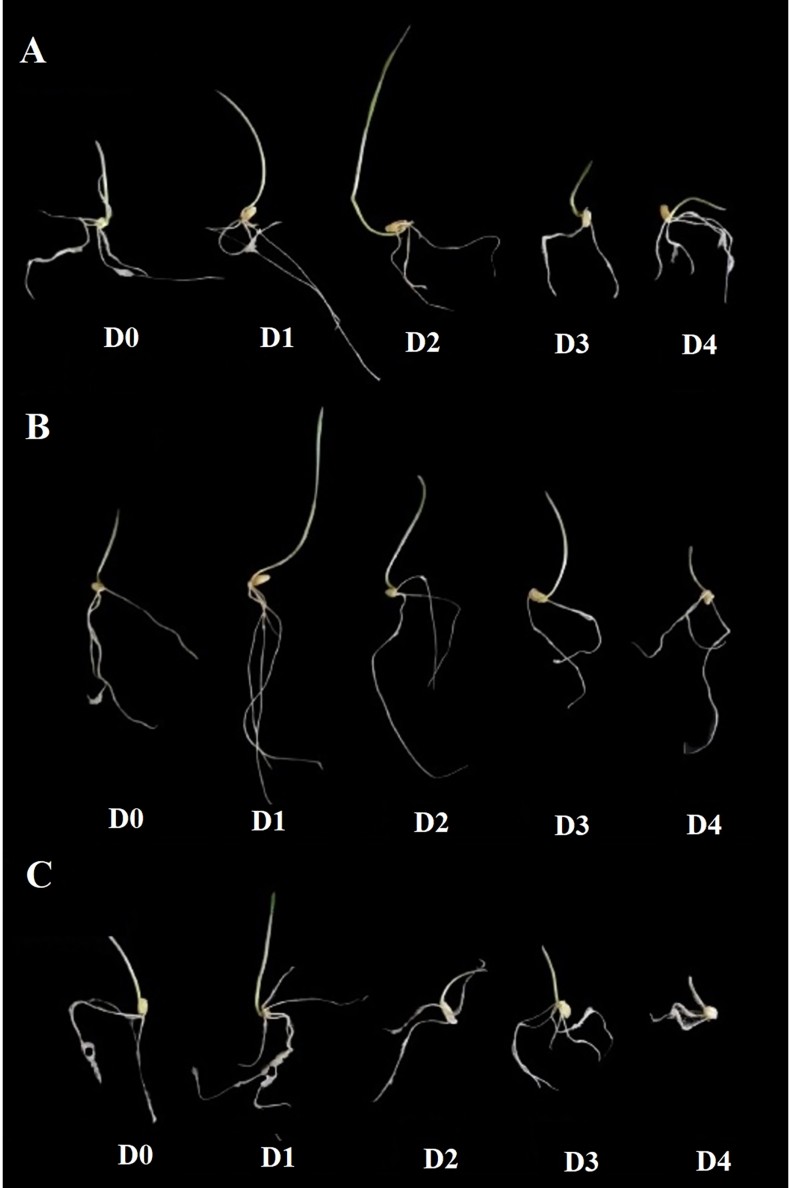

**Figure 4 The effect of essential oil types and application doses on coleoptile length.** The effect of rosemary essential oil on coleoptile length (A), the effect of sage essential oil (B), and the effect of lavender essential oil (C) were stated. The application doses represent D0: control, D1: 0.01%, D2: 0.05%, D3: 0.10%, D4: 0.25%.               

The highest total chlorophyll value was determined as 516.60 µg/g in the $D_3$ treatment dose (Fig. 5C). Compared to the $D_0$ control treatment, other essential oil doses had a negative effect on the total chlorophyll content. On the other hand, rosemary essential oil is the most effective on total chlorophyll value among the other essential oil types (Fig. 6C).

## Field experiments results

As a result of the analysis of the field soil, the total nitrogen (N) content is 0.11%, the phosphorus ($P2O_5$) content is 7.93 kg/da, the potassium ($K_2O$) content is 113.1 kg/da, and

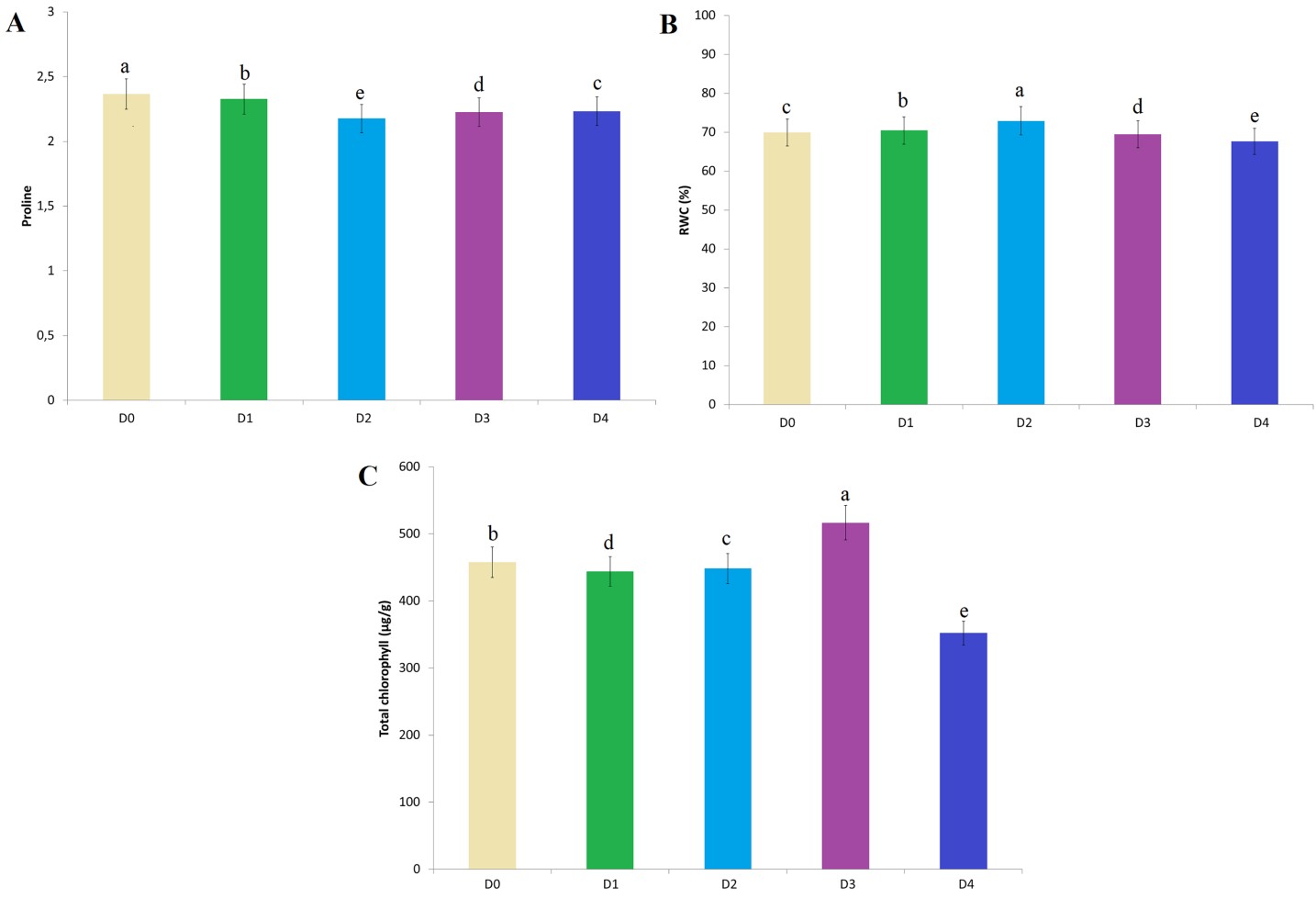

**Figure 5 The effect of essential oil treatment doses on the amount of proline, RWC and total chlorophyll content.** (A) Proline content of the plant. (B) RWC of the plant. (C) Total chlorophyll content. Treatment doses represent D0: control, D1: 0.01%, D2: 0.05%, D3: 0.10%, D4: 0.25%. The difference between the means shown with different letters is significant according to the Duncan test at the 0.01 level.

the calcium ($CaCO_3$) content is 9.25%. Besides, the organic compound is 0.99%, the total organic carbon content is 0.77%, the salt (NaCl) content is 0.14%, the pH is 7.98 and the electrical conductivity (EC) was determined as 1.7 $dSm^1$. According to the results of the soil analysis, the nitrogen and organic matter content of the soil is insufficient. Organic carbon content is also low. Phosphorus and potassium content is moderate, salt content is harmless for plants. It shows alkaline properties according to the soil pH value. The soil EC value is medium, indicating that the soil is slightly salty. Soil lime content was determined at the medium calcareous—calcareous level.

According to the meteorological data of the field (Fig. 1), it was observed that the average temperature in October, when wheat was planted in the 2019–2020 period, was below 20 °C and the precipitation was insufficient until the end of 2019. It has been determined that the amount of precipitation is below seasonal normal throughout the year, despite the temperatures continuing at seasonal normal in 2019–2020. Especially in 2020,

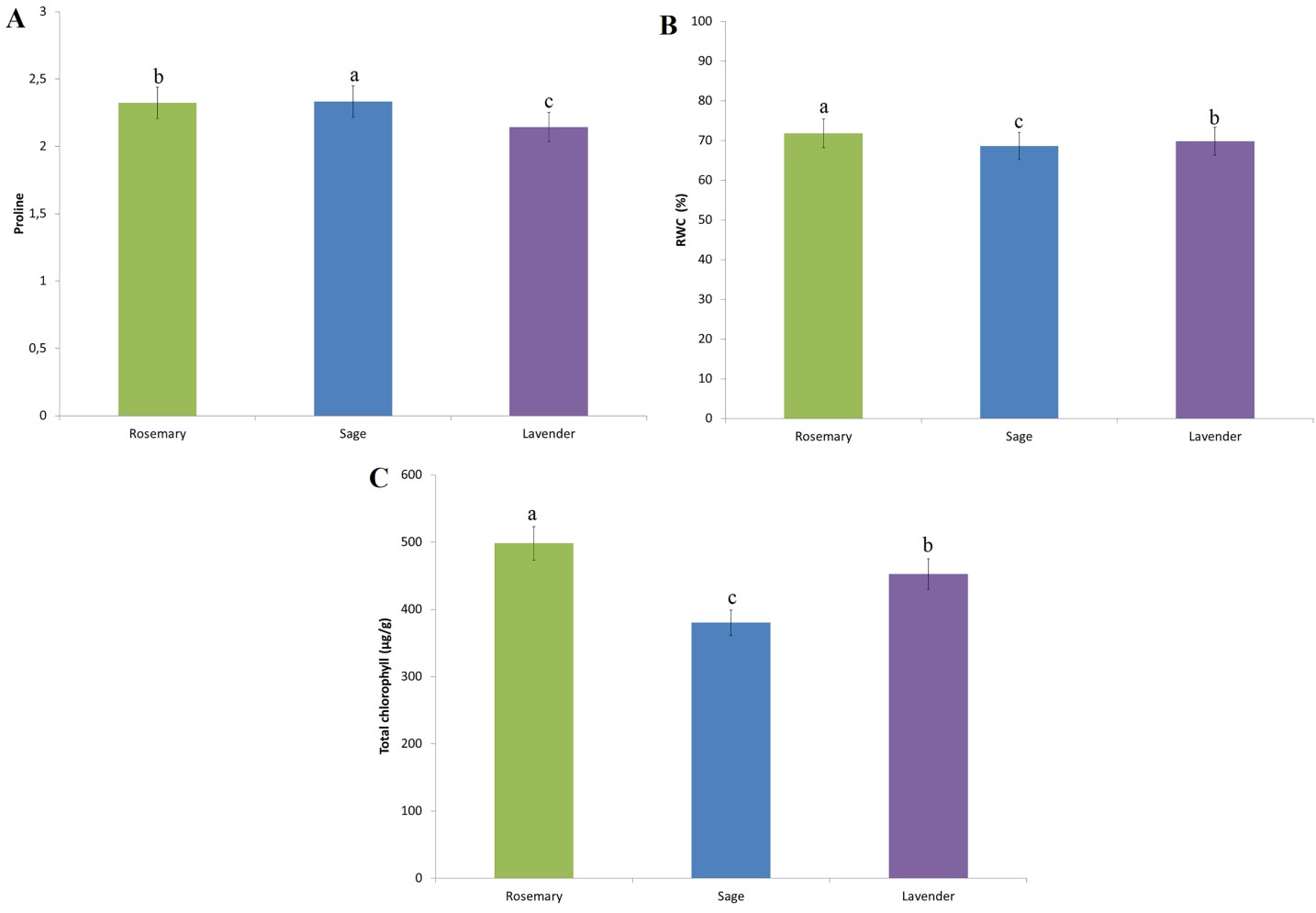

**Figure 6 The effect of essential oil species on the amount of proline, RWC and total chlorophyll content.** (A) Proline content of the plant. (B) RWC of the plant. (C) Total chlorophyll content. Treatment doses represent D0: control, D1: 0.01%, D2: 0.05%, D3: 0.10%, D4: 0.25%.The difference between the means shown with different letters is significant according to the Duncan test at the 0.01 level.

it was a dry year with less precipitation. Long-year average precipitation forecast data for Ankara is given in Fig. S1A. The values indicated in the table show that the precipitation has been below the seasonal normal in recent years, with a decreasing trend. The long-term average temperature data of Ankara is given in Fig. S1B. The values indicated in the table show that the temperature has been above the seasonal normal in recent years with an increasing trend.

According to obtained field experiment results, the highest value in terms of the number of plants per square meter was determined in the seeds covered with rosemary essential oil (445 plants) (Table 2). There is no difference between the number of plants per square meter determined in lavender treatment and control (429 and 430 plants, respectively). Plant height was determined as 99.45 cm in the control treatment during the harvest period. This value decreased with other essential oil treatments. Average plant height was measured as 91.86 cm in the lavender treatment and 84.87 cm in the rosemary treatment.

**Table 2 The effects of essential oil types on yield and growing parameters in seed biopriming treatment.**

| Essential oil type | PN (m²/plant) | PH (cm) | SL (cm) | SGN | SGY (g) | GY (kg/da) | TGW (g) |
|---|---|---|---|---|---|---|---|
| *Rosmarinus officinalis* | 445[a] | 84.87[c] | 9.30[ns] | 16.90[ns] | 7.50[ab] | 256.52[a] | 43.30[a] |
| *Salvia officinalis* | 431[b] | 80.79[d] | 9.10[ns] | 17.90[ns] | 8.60[a] | 234.36[b] | 40.14[b] |
| *Lavandula intermedia* | 429[c] | 91.86[b] | 9.55[ns] | 18.90[ns] | 6.50[b] | 224.62[c] | 39.58[b] |
| Control | 430[bc] | 99.45[a] | 9.45[ns] | 20.70d[ns] | 6.40[b] | 219.34[d] | 36.09[c] |

Note:
The difference between the means shown with different letters in the same column is significant at the 0.01 level. Ns, non-significance differences; PN, Number of plants per square meter; PH, Plant height at harvest period; SL, Spike length; SGN, Number of grains per spike; SGY, Spike grain yield; GY, Grain yield per unit area; TGW, Thousand-grain weight.

Especially the lowest plant height was measured as 84.87 cm in sage essential oil treatment (Table 2). The difference between treatments in spike length was statistically insignificant (Table 2).The number of grains per spike was determined as 20.70 in the control treatment. The average number of grains per spike was determined as 18.90 in the lavender treatment, 17.90 in the sage treatment and 16.90 in the rosemary treatment. However, the difference between them was found to be insignificant. The difference between treatments on spike length is statistically insignificant (Table 2). The most positive effect on the number of grains in the spike was determined in the sage treatment (21.70 pieces). The number of grains in the spike was determined in rosemary and lavender treatments lower than in the control (Table 2).

Spike grain yield was determined as 6.40 g in the control treatment. There is no statistical difference between the control and the lavender essential oil treatment (6.50 g). On the other hand, the most significant value compared to the control was determined in the sage treatment (8.60 g). In sage and rosemary treatment, no difference was determined in terms of grain yield values (8.60 and 7.50 g, respectively) (Table 2). The most effective treatment on grain yield was determined as rosemary (256.52 kg/da). In addition, the grain yield values determined in sage and lavender treatments are important compared to the control treatment (234.36, 224.62 and 219.34 kg/da, respectively) (Table 2). The most important effective treatment on thousand seed weight was determined as rosemary (43.30 g). The difference between sage and lavender treatments was determined statistically insignificant (40.14 and 39.50 g, respectively). The lowest thousand-grain weight was determined as 36.09 g in the control treatment (Table 2).

# DISCUSSION

Arid and semi-arid environmental conditions induce the formation of reactive oxygen species (ROS) in plants and cause oxidative damage to plant cells. ROS signaling is involved in the initiation of stress-induced molecular, biochemical, physiological and morphological responses (*Sharma et al., 2021*). High levels of ROS production can damage various physiological and metabolic processes in plants, such as stomatal activity, osmotic adjustment, RWC, chlorophyll content, photosynthesis, RWC, and antioxidant defense system (*Oguz et al., 2022*). Antioxidant system and osmotic regulation are the basic defense systems that provide tolerance to water deficiency stress conditions in plants (*Zou, Wu &*

*Kuča, 2021*). However, the seed germination process is regulated by ROS (*Bailly, 2019*). The produced ROS regulate gene expression and phytohormones signaling and homeostasis of abscisic acid, gibberellins, auxins and ethylene to control cellular events related to seed germination (*Choudhary, Kumar & Kaur, 2020*). Low ROS levels have a positive effect on germination, while high ROS causes oxidative damage that inhibits seed germination (*Bailly, 2019*). The increase in germination induced by seed priming is due to the low level of ROS accumulation (*Hussain et al., 2019*; *Nile et al., 2022*). Besides, priming facilitates water uptake into expanding tissues as a result of the increase in the expression of aquaporin genes, which play a vital role in plant-water relations (*Raj & Raj, 2019*). In our study, increased germination rate with essential oil priming was associated with low-dose ROS accumulation and increased activation of genes involved in water uptake.

Positive effects on germination with seed priming methods can be evaluated under different hypotheses. The first is based on the fact that seed preparation causes an increase in energy metabolism, and takes an active role in the consumption of seed nutrient stores and in the elongation of the embryo (*Chen & Arora, 2011*). The second hypothesis is that seed priming creates a "stress memory" in germinating seeds by imposing stress conditions on seeds that suppress seed germination but induce enzyme activation and osmotic adjustment (*Ibrahim, 2016*). This hypothesis is based on the fact that seed preparation provides plant growth and stress tolerance by increasing the accumulation of osmotic substances such as proline, soluble sugars and soluble proteins, which regulate plant water potential, and by contributing to the regulation of protective activities of enzymes such as catalase (CAT), superoxide dismutase (SOD) and peroxidase (POD) (*Pal, Ali & Pal, 2017*). Essential oils treatment in the seed priming method can increase or suppress the activity of free radical scavenging enzymes such as SOD, CAT and POD in the plant. *Kubala et al. (2015)* reported an increase in CAT enzyme during germination with seed preparation. For this reason, it was emphasized that the antioxidant defense system should be stimulated for seedling formation.

It has been reported by many researchers that phenolic compounds and plant extracts have biostimulant effects on seed germination, rooting and shoot development by treated essential oils to seeds, leaves or soil (*Kisiriko et al., 2021*). This bioactivity is known to result from the interaction between the different components of essential oils. Organic compounds of essential oils obtained from aromatic plants show antioxidant activity. This effect is aimed at protecting the tissue from oxidative stress by preventing the degradation of auxin in the plant. As a result, phenolic compounds in organic form protect the plant from stress and balance the hormone activity, and improve physiological activities such as plant height, root length and germination (*De Klerk et al., 2011*).

Essential oils can inhibit germination with secondary metabolites such as terpenes and phenolic compounds they contain. This allelopathic effect of essential oils is closely related to their concentration (*Wang et al., 2017*). *Bingöl & Battal (2017)* reported that purslane and corn seeds treated with extract prepared using *Salvia limbata* essential oil showed different effects on germination rates. It was stated that while the germination percentage in purslane seeds was 80.7% at 3% extract concentration, it decreased to 23.8% at 5% concentration and germination did not occur at increasing concentrations (7% and 9%).

Similarly, it was reported that germination decreased significantly after 7% essential oil was applied to corn seeds. *Binbir et al. (2019)* stated that the germination rate of corn seeds decreased with increasing essential oil dose, depending on the ratio of essential oil obtained from lavender. On the other hand, *Săndulescu, Manole & Stavrescu-Bedıvan (2020)* reported that low-dose essential oil treatment did not cause any negative effects on the germination of tomato seeds. According to the results of the study, it was determined that in treatments containing low doses of essential oil ($D_1$ and $D_2$), although it differed according to the type of essential oil, germination rates increased compared to the control. On the other hand, it was determined that increasing essential oil contents ($D_3$ and $D_4$) showed a suppressive effect on wheat germination rates (Fig. 2A). The main reason for the decrease in germination, at increasing essential oil treatment doses, can be associated with high levels of ROS accumulation in the seed.

*Atak, Mavi & Uremis (2016)* reported that essential oils obtained from *Origanum onites* L. and *Rosmarinus officinalis* L. species had a positive effect on the germination of wheat seeds, but had a negative effect on shoot & root length and fresh weight. It has been reported that this difference in effect is due to the allelopathic effect of the essential oil. In addition to the treatment dose, the type of EO plant, active ingredient content and texture may have different phytotoxic effects (*Zahed et al., 2010*). Besides, the response of the plant to the essential oil treatments is closely related to the species and variety characteristics of the treated plant. *Day (2016)* stated that the essential oils obtained from the stem and root tissues of the safflower plant showed different effects on the germination rates of wheat, barley, sunflower and chickpea. In the current study, sage essential oil showed a better effect than rosemary and lavender. The different effects of essential oils on the germination rate of wheat seeds are closely related to the content of essential oils.

Coleoptile length is of great importance for the germination rate and seedling development of the planted seeds in the field. The shoot length and plant height of the wheat genotypes with a short coleoptile length is also short. It has been stated that as the length of the coleoptile increases, the plants that have the possibility of photosynthesis for a longer time thanks to the rapid seedling formation and leaf formation are more advantageous against environmental stresses and yield losses (*Na et al., 2009*). *Murphy et al. (2008)* reported that the length of the coleoptile varies between 59–159 mm, although it differs according to the wheat genotypes. Moreover, the length of the coleoptile showed a positive and significant relationship with the plant height. According to *Rebetzke et al. (2007)* thanks to the long coleoptile property of wheat genotypes, increased germination and emergence, seedling formation rate, and achieved more plant growth per unit area. Besides, it has emphasized that thanks to the high coleoptile, high number of siblings, leaf area and grain yield were obtained. According the current study, it was determined that the length of the coleoptile can be increased significantly with essential oil treatment. Although it varies according to the essential oil type, significant positive increases in coleoptile lengths were determined in $D_1$ and $D_2$ treatments compared to control (Figs. 2B, 4). On the other hand, $D_3$ and $D_4$ treatments showed a suppressive effect on coleoptile lengths as well as germination rate.

Organic compounds contained in essential oils obtained from aromatic plants interact with the endogenous hormone content of the plant. Organic compounds balance the hormone level by preventing the degradation of the plant's endogenous auxin (*De Klerk et al., 2011*). In this way, it has a positive effect on physiological activities such as germination, plant height and root length. *da Silva, Dobranszki & Ross (2013)* stated that polyphenol derivatives, one of the phenolic compounds found in the essential oils of medicinal aromatic plants, support shoot and root formation and development under *in vitro* conditions. However, it was emphasized that the applied polyphenol derivative compounds showed inhibitory effects on growth at high doses. According to the results of the current study, $D_1$ and $D_2$ treatment doses showed the best positive effect on the shoot length (mean 17.90 and 18.51 cm, respectively). Although there is no statistical difference between $D_1$ and $D_2$ treatments; showed a significant positive effect relative to the control (Fig. 2C). Shoot lengths determined at the $D_4$ treatment dose were significantly lower than the $D_0$ treatment in all essential oil types. Similarly, $D_4$ essential oil treatments showed a suppressive effect on root lengths (Fig. 2D). According to the control $D_0$ treatment, the best root length was measured as a mean of 13.76 cm in the $D_2$ treatment. There is no significant difference between the effects of $D_1$ and $D_3$ treatments on root lengths (mean 12.63 and 12.65 cm, respectively). In addition, the effects of essential oil types on root length were statistically in the form of sage, rosemary and lavender, respectively.

In drought-induced oxidative stress osmotic adjustment in plants occurs with the accumulation of low molecular weight organic solutions (*Marcińska et al., 2013*; *Rao & Chaitanya, 2016*). These organic solutions are found in plants as soluble carbohydrates and proline. Increasing proline accumulation in the plant is effective to reduce damage under water stress conditions (*Anjum et al., 2011*). Proline is associated with osmotic regulation, membrane stabilization, and water content regulation in plants (*Hayat et al., 2012*; *Zadehbagheri, Azarpanah & Javanmardi, 2014*). Proline shows high antioxidant properties and plays a major role in the prevention of cell death (*Jyoti & Yadav, 2012*). In addition, stress-induced proline accumulation of plants is important in determining the stress tolerance capacity of the plant (*Saeedipour, 2013*; *Mwadzingeni et al., 2016*). In this study, the decreased amounts of proline with essential oil treatments may be due to the interaction of the plant with stress tolerance mechanisms in a stress-free environment. On the other hand, there is a significant relationship between low proline content and high RWC in a stress-free controlled environment. The lowest proline accumulation was determined in the $D_2$ treatment, in which the highest RWC content was determined (Figs. 5A and 5B). This effect supports the idea that the treatment of essential oil together with the antioxidant content of the plant will be effective in increasing stress tolerance. On the other hand, among essential oil types, rosemary and sage essential oils showed a more significant effect on proline accumulation than lavender.

Leaf water content is one of the important parameters measured by many researchers in the determination of drought-resistant cultivars and breeding studies (*Ahmad et al., 2022*). Although drought stress tolerance is known to be associated with high water potential in tissues, the water potential of stressed plants is lower than that of plants in a non-stressed environment (*May & Milthorpe, 1962*; *Eastham, Oosterius & Walker, 1984*). In addition,

drought stress affects physiological and chemical properties such as leaf area, chlorophyll content, and leaf length. In the study, the highest RWC values were measured in the $D_2$ treatment compared to the $D_0$ treatment (69.85% and 72.90% respectively) (Fig. 5B). *Binbir et al. (2019)* reported that the germination rate decreased and the dry weight of the seedling increased as a result of the treatment of lavender essential oil to corn seeds. Besides, as the essential oil dose increased, moisture content, root length, seedling length and seedling fresh weight also decreased. According to the current study results, $D_3$ and $D_4$ treatment doses have a negative effect on RWC (Fig. 5B). *Chandrasekar, Sairam & Srivastava (2000)* stated that drought stress caused a decrease in the relative water content and chlorophyll amount in all genotypes in their experiment, while it caused an increase in the accumulation of proline and abscisic acid (ABA). *Ben-Jabeur et al. (2019)* reported that coating wheat seeds with thyme oil were successful in reducing the accumulation of ABA under water stress and minimizing the reduction in the amount of water available in the plant. This improvement is due to the interrelationship of a series of biochemical, molecular, cellular and physiological events. The positive effect of essential oils on leaf water content reveals the significant potential of essential oil treatments in stress conditions (*Farooq et al., 2009*).

The increased amount of chlorophyll with essential oil treatments supports the view that some mechanisms are activated to reduce the damage caused by drought (*Basu et al., 2016*). However, the reduction of chlorophyll damage with essential oil treatments also supports the idea that photosynthetic capacity is preserved and the plant can maintain its vital functions by reducing oxidation damage (*Hayat et al., 2012*). The variable effect of essential oil treatments on total chlorophyll content in our study can be attributed to both of the above-mentioned situations. However, the chlorophyll contents varying according to the control treatment at each treatment dose ($D_1$, $D_2$, $D_3$ and $D_4$) will contribute to the accepted idea that essential oil treatments stimulate the plant's stress tolerance mechanism (Fig. 5C).

In general, rosemary and sage essential oil treatments have a more positive effect than lavender. This situation is largely related to essential oil components. A high rate of 1.8-cineole was determined in the essential oil content of rosemary and sage (35.8% and 16.67%, respectively) (Table 1). On the other hand, a high rate of 28.64% linalool content determined in lavender may have had a suppressive effect on the parameters studied. However, the effect of essential oil on seed germination and seedling establishment should be considered the collective effect of essential oil contents.

Plant seed variety and environmental factors are of great importance to the number of plants/$m^2$ (*Lloveras et al., 2004*). *Dinç (2010)* reported that if the planting frequency is above 500 plants/$m^2$, the plant height decreases. Besides, the effect of different plant densities on grain yield was not significant. According to the results of the study, although the number of plants/$m^2$ showed a statistical difference, it was accepted that the effects caused by the number of plants per square meter disappeared due to the appropriate planting density. *Aktaş (2010)* reported the average plant height of Köse wheat as 101.9 cm in the study they conducted in Ankara under dry conditions. In the same study, the spike length of Köse wheat was 9.9 cm, the grain yield per unit area was 182.8 kg/da, and the

thousand-grain weight was 32.37 g. These results compared to the current study, it has been determined to have a significant positive effect on the yield and yield parameters of seed priming with essential oils. As a result of our study, the yield was calculated in control 219.34, 256.52 kg/da in rosemary, 234.36 kg/da in sage and 224.62 kg/da in the lavender treatment. Additionally, the spike grain yield and thousand-grain weight increased with rosemary essential oil treatment. *Ben-Jabeur et al. (2022)* reported that, seed coating technique with thyme oil resulted in improvement in vegetative growth, grain yield and agronomic components (spike/m$^2$, straw yield, thousand-grain weight and harvest index) under drought stress. The increases in yield were associated with the regulation of ABA in plants treated with thyme oil (*Ben-Jabeur et al., 2019*). According to the results of the studies based on gene expression analysis, essential oil treatment play an important role in the activation of molecular mechanisms in tolerance against abiotic and biotic stress factors (*Ben-Jabeur et al., 2015*; *Banani et al., 2018*; *Sukegawa et al., 2018*; *Kesraoui et al., 2022*). Essential oil treatment is effective in activating JA, ET and SA biosynthesis and their hormonal interactions including complex signaling steps (*Rienth et al., 2019*). The increase in yield achieved by seed priming with essential oil in the current study can be attributed to the multiple collective actions of different mechanisms active at the molecular level. Consequently, the obtained study findings support the idea that the increase observed in crop productivity is due to the bio-regulatory and biostimulant effects of essential oil treatments.

Different seed preparation methods are known to increase plant performance effectively (*do Espirito Santo Pereira et al., 2021*). Considering the 20–30% loss in wheat yield, especially with the effect of drought (*Zhang et al., 2018*); the aim of the seed priming studies is to maintain wheat yield in an environmentally and economical way. The use of plant extracts and aromatic oils in agriculture has been proposed as an environmentally friendly and cost-effective approach (*Farooq et al., 2018*; *Ben-Jabeur et al., 2022*). According to the meta-analysis results performed by *Mickky (2022)*, different priming techniques caused an important increase in wheat economic yield (29%), biological yield (22%) and thousand-grain yield (16%). It was also stated that these studies showed high consistency with each other. Compared to chemical inputs, extracts and essential oils obtained from plant parts allow inexpensive and easily reproducible applications. Besides, seed preparation methods should be tried and tested in different field conditions, climates and regions to contribute to the success of priming and meet the needs of farmers (*El Boukhari et al., 2020*; *Ben-Jabeur et al., 2022*). In the present study, successful results were obtained that will contribute to the importance of the seed priming method with essential oil in increasing wheat yield performance in arid and semi-arid regions.

## CONCLUSIONS

According to the study findings, an increase was observed in the physiological and biochemical parameters of the plant with priming. Significant differences were determined in the parameters related to the yield in the field condition. These differences resulted in accordance with the proposal of the study. Treatment dose optimization was carried out in the use of essential oils as a bio-stimulator. Important findings have been obtained

regarding the applicability of essential oils and priming in field conditions and in sustainable farming systems. The findings will contribute significantly to the studies to be carried out on this subject. Besides, it is possible to obtain important results in the experiments to be carried out with the chemical derivatives of these EO components in order to determine the effect of the single component of the EO on plant development. In addition to the obtained data, there is a need for a detailed examination of this subject with molecular, physiological and methodological treatments. Important results will be obtained in the evaluation of field trials in different regions and climates. It is recommended to study priming treatments at different doses and times, especially in field trials.

## ACKNOWLEDGEMENTS

We thank Dr. Yasin ÖZGEN from the Ankara University Faculty of Agriculture Department of Field Crops, for assisting in the supply of aromatic plants, the extraction of essential oils, and the analysis of essential oil components.

### Funding

The authors received no funding for this work.

### Competing Interests

The authors declare that they have no competing interests.

### Author Contributions

- Muhammet Çağrı Oğuz conceived and designed the experiments, performed the experiments, analyzed the data, prepared figures and/or tables, authored or reviewed drafts of the article, and approved the final draft.
- Ezgi Oğuz analyzed the data, prepared figures and/or tables, authored or reviewed drafts of the article, and approved the final draft.
- Mustafa Güler conceived and designed the experiments, authored or reviewed drafts of the article, and approved the final draft.

### Data Availability

The raw data and measurements are available in the Supplemental Files.

### Supplemental Information

Supplemental information for this article can be found online at http://dx.doi.org/10.7717/peerj.15126#supplemental-information.

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
