# Peer review of "Seed priming with essential oils for sustainable wheat agriculture in semi-arid region"

_PeerJ, doi:10.7717/peerj.15126_

## Round 0.1 · original submission · Major Revisions

Dear authors:
This manuscript is an interesting and potentially important topic, and the standard of presentation of the research is good. However, there are many notes that should be considered before accepting this manuscript, as follows:

- The title should represent the article's content and facilitate retrieval in indices developed by secondary literature services. A good title (i) briefly identifies the subject, (ii) indicates the purpose of the study, and (iii) gives important and high-impact words early.
- The abstract must be completely self-explanatory and intelligible in itself. It should include the following: 1. Reason for doing work, including rationale or justification for the research; 2. Objectives and topics covered; 3. Brief description of methods used; 4. Results; 5. Conclusions.
- M&M: No sufficient information was provided to describe both the experimental procedures of laboratory and field experiments. The experimental design in both experiments was not mentioned. So, it is needed to define the experimental design and also clarify how authors statistically analysed both experiments.
- Results: The legends of each figure should be improved and mentioning which multicomparison test used to distinguished between treatments and at each level 0.05 or 0.01 probability. Table 2: authors need to provide footnote about the meaning of small letters followed the mean values of each treatment for each trait.
- Discussion: Discussion needs to be rewritten and improved with adding recent citations.

Reviewer 1 ·

Basic reporting

- The title (label) of figures must be under it.

Experimental design

- The methods described with sufficient detail to lesser extent.
- It was possible to add the endogenous hormone analysis to enhance the results and discuss them, especially since the author cited the correlation of IAA and ABA and linked them in the discussion of the results (lines 298, 398 and 401).

Validity of the findings

No comment

Additional comments

I thank you for providing the raw data. It’s clear, well organized and rationality.
I commend the authors for their extensive data set, compiled over many years of detailed fieldwork. In addition, the manuscript is clearly written in professional, unambiguous language. If there is a weakness, it is in the title of figures which be improved.

Annotated reviews are not available for download in order to protect the identity of reviewers who chose to remain anonymous.

·

Basic reporting

The aim and the idea of the manuscript is very good but there is some correction in it to improve the manuscript

Experimental design

Good

Validity of the findings

good

Additional comments

The discussion need to improvement

Reviewer 3 ·

Basic reporting

The aim of this research is to measure the effects of Rosmarinus officinalis, Salvia officinalis and 27 Lavandula x intermedia essential oils doses of %0.01, %0.05, %0.10 and %0.25 as a bio28 stimulator on wheat biochemical and physiological parameters in the laboratory. However, the following suggestions are recommended:
• Your abstract does not highlight the specifics of your research or findings. Furthermore, abstract is written in smaller subsections. This is not a proper way of writing abstract of any research paper. Present whole abstract in one paragraph.
• In the introduction section I suggest: problems, Aim, Methods, Results, and Conclusion. The author needs to explain the major factor of the manuscript. The paper's aim has been evidenced very poor, it is strongly suggested to highlight the originality and added value of the present work with respect to the Literature about the same topic. Introduction suffers from a lack of motivation and innovations. It should be expanded to include a more detailed discussion of current problems.
• The contribution section is missing at the end of the introduction section. Add a contribution paragraph as a second last paragraph in the introduction section.
• The paper organization section is missing at the end of the introduction of section. Briefly describe the section and subsection of your whole menu script in one paragraph. Add this paragraph at the end of the introduction section.
• Add a literature review section after introduction section.
• There should be no consecutive headings, add some text between two headings.
• Results and Discussion; the author should compare the finding of the present study with the previous study and justify for more clarity.
• Would you explicitly specify the novelty of your work? What progress against the most recent state-of-the-art similar studies was made?
• Conclusions should be amended to incorporate a broader discussion of the significance and potential application of this specific study.
• English throughout the manuscript needs to be improved.
• I will not recommend this article for publication in this journal. Resubmission is allowed after major revision.

Experimental design

no comment

Validity of the findings

no comment

Additional comments

Paper can be accepted after addressing of suggested comments

·

Basic reporting

Title: Seed priming with essential oils for sustainable wheat agriculture in semi-arid region.
Abstract:
Line 14: “The biggest impact of the disturbing nature balance on the world is global climate change.” Rewrite it.
A standards (internationally) abstract should have numerical &/ statistical inputs. Do the needful.
Line 27: use the % properly.
Line 36-38: it seems the claim is justified to some extent. However, impacts of essential oil as biopesticides etc. are known.
Kesraoui, S., Andrés, M.F., Berrocal-Lobo, M., Soudani, S. and Gonzalez-Coloma, A., 2022. Direct and Indirect Effects of Essential Oils for Sustainable Crop Protection. Plants, 11(16), p.2144.

Introduction: ok.
Authors are advised to include drought induced ROS mediated adverse effects on crops, highlighting the selected crops.
I will suggest the authors to include an hypothesis.

Experimental design

Materials & Methods
Plant materials: Here species/ varietal profiles are to be included. Do the needful.

Preparation of essential oil solutions: Authors are advised to present a pictorial (comparison) with / without Tween 20 to ensure that oils were completely dissolved not forming any emulsion.

Seed sterilization and priming treatments: Here suitable references are required.
Design and treatment combinations need to be disclosed.

Statistical Analysis: Rewrite this section with details.

Validity of the findings

Results:Seems to be ok, minor English language issues can be resolved.

Discussion: here, correlation with ROS will boost up the quality of MS. Integration of findings will be an asset.

Why D2 onward there is a downwards trend is observed. Add suitable justification.
Figure 2 and on: what does the different letter cases indicates? Rewrite all properly.
You can enhance the contrast of the figures.
## Authors are required to add a paragraph on coast beneficial analysis of the outcome and how the farmer community will be benefited.

In my opinion, the authors have presented an interesting finding with novel approach. However, the MS have some issues, need to be resolved. I am suggesting a MAJOR REVISION.

---

## Round 0.2 · Minor Revisions

Dear authors:
Please respond to my comments below and comments of Reviewer #2

- The title should represent the article's content and facilitate retrieval in indices developed by secondary literature services. A good title (i) briefly identifies the subject, (ii) indicates the purpose of the study, and (iii) gives important and high-impact words early.

- The abstract must be completely self-explanatory and intelligible in itself. It should include the following: 1. Reason for doing work, including rationale or justification for the research; 2. Objectives and topics covered; 3. Brief description of methods used; 4. Results; 5. Conclusions.

- M&M: No sufficient information was provided to describe both the experimental procedures of laboratory and field experiments. The experimental design in both experiments was not mentioned. So, it is needed to define the experimental design and also clarify how authors statistically analysed both experiments.

- Results: The legends of each figure should be improved and mentioning which multicomparison test used to distinguished between treatments and at each level 0.05 or 0.01 probability. Table 2: authors need to provide footnote about the meaning of small letters followed the mean values of each treatment for each trait.

- Discussion: The discussion needs to be rewritten and improved by adding recent citations.

Reviewer 1 ·

Basic reporting

The author rewrite several parts of the manuscript as abstract, methods, conclusion and several paragraphs in the introduction and discussion to be more accurate and suitable for publication in this journal.

Experimental design

No comment

Validity of the findings

No comment

Additional comments

The manuscript become more valuable to be published in this journal

·

Basic reporting

Seed priming with essential oils for sustainable wheat agriculture in semi-arid region (#77380)

Line 34& 35 it can be deleted (The light of these results, 35 the effects of essential oil types and doses on yield parameters were discussed).
Materials & Methods
Line 110 you can use tolerant instead of resistant
Line 134 you can write The control treatment was prepared instead of It was dilute
Line 137 where is Figure S2
Line 142 you can delete (sodium hypochlorite)
Line 148 delete one of the two dots (.)
Lines 157-161 review the RWC with the reference
Statistical analysis need reference
Lines 502-510 can be transferred to the pegining of the discussion

Experimental design

review the RWC with its reference

Validity of the findings

Good

Additional comments

The text need to formatting

·

Basic reporting

.

Experimental design

.

Validity of the findings

.

Additional comments

It seems to me that the authors have answered the issues raised by the reviewers in a more or less satisfactory manner. Improvement in MS can be seen.
The editor and authors are advised to take care of minor details of the MS.
I am suggesting ACCEPT of R1 version.

---

## Round 0.3 · Minor Revisions

Dear Authors,

Although the manuscript has been improved, it is still you did not provide the experimental design for each experiment and how you statistically analyzed your experiments. For example, if you have a single factor that means you have to design your experiment with either RCBD or CRD, and so on.

Best regards.
Magdi Abdelhamid

---

## Round 0.4 · Minor Revisions

Dear authors,

I have only one issue. Please move the experimental design description of both experiments from the Statistical analysis section into the Field trial and the experimental design section. Any revisions to the manuscript should be marked up using the highlighting function if you are using MS Word, such that any changes can be easily viewed by the editors and reviewers.

Best regards

Magdi Abdelhamid

---

## Round 0.5 · accepted · Accept

Thank you for revision the manuscript